# Knowledge Graph in Astronomical Research with Large Language Models: Quantifying Driving Forces in Interdisciplinary Scientific Discovery

## Abstract

Identifying and predicting the factors that contribute to the success of interdisciplinary research is crucial for advancing scientific discovery. However, there is a significant lack of methods to quantify the integration of new ideas and technological advancements within a field and how they trigger further scientific breakthroughs. Large language models, with their prowess in extracting key concepts from vast literature beyond keyword searches, provide a new tool to quantify such processes. In this study, we use astronomy as a case study to quantify this process. We extract concepts in astronomical research from 297,807 publications between 1993 and 2024 using large language models, resulting in a refined set of 24,939 concepts. These concepts are then adopted to form a knowledge graph, where the link strength between any two concepts is determined by their relevance based on the citation-reference relationships. By calculating this relevance across different time periods, we quantify the impact of numerical simulations and artificial intelligence on astronomical research, demonstrating the possibility of quantifying the gradual integration of interdisciplinary research and its further branching that leads to the flourishing of scientific domains.

## 1 Introduction

Interdisciplinary collaborations often drive innovation in research by introducing new theoretical, analytical, or computational tools to specific scientific domains. These new tools can revitalize and open up fields that might otherwise remain stagnant. For instance, the theoretical understanding of quantum physics and general relativity has driven much of modern cosmology [Weinberg, 2008], and each subsequent engineering breakthrough leads to new windows of observation. A prime example is the detection of gravitational waves with LIGO [Abbott *et al.*, 2016], which was made possible by the convergence of cutting-edge technologies in interferometry. Simultaneously, high-performance computing has paved the way for understanding complex systems in the cosmos, such as the evolution of galaxies [McAlpine *et al.*, 2016;

Pillepich *et al.*, 2018] and the inner workings of stars and stellar atmospheres [Gudiksen *et al.*, 2011], through N-body or hydrodynamical simulations.

The advancement of astronomy also relies heavily on the revolution of statistical and analytical methods, which allow for proper inferences based on observations. The introduction of even well-known statistical techniques to astrophysics often leads to key turning points in the field. For example, a cornerstone of our understanding of cosmology comes from analyzing the power spectrum of the cosmic microwave background [Hu and Dodelson, 2002], while the detection of planetary systems outside the solar system has benefited from Gaussian Processes [Hara and Ford, 2023]. More recently, the advent of deep learning, with numerous successes in sciences such as AlphaFold [Jumper *et al.*, 2021], has propelled much of the field to rethink statistical inference in astronomy. This includes using generative models as surrogates for the likelihood or posterior [Cranmer *et al.*, 2020; Sun *et al.*, 2023a] and employing flow-based generative models to capture higher-order moment information in stochastic fields [Diaz Rivero and Dvorkin, 2020].

However, the underpinnings of these successful interdisciplinary results often stem from a rigorous process of debate and adaptation within the community. New thought processes are initially treated as disruptors, but a subset of these promising methods subsequently becomes integrated into the field's knowledge base. Over time, such integration gains significant traction and further creates branching of knowledge in the field, fostering its growth. Consider the example of numerical simulation, which was initially viewed as a "distraction" from pure mathematical interest in solving N-body problems and Navier-Stokes equations [Bertschinger, 1998]. However, astrophysics has gradually acknowledged that some aspects of the field are nonlinear and beyond analytical understanding. The integration of numerical simulations has subsequently led to the thriving study of galaxy evolution [McAlpine *et al.*, 2016], a widely researched topic, and has also gradually permeated into more specialized domains like solving the accretion physics of black holes and protoplanetary disks [Jiang *et al.*, 2014; Bai, 2016].

However, while such integration and branching off are intuitively clear, studying and quantifying them remains a challenge. Questions such as how long it might take for a field

to adopt a new concept and what quantitative impact it has on the field still evades rigorous study. A key bottleneck is the difficulty in defining and extracting the various concepts described in a paper. The classical approach of classification using only keywords or the field [Xu *et al.*, 2018] of research might lack granularity. Other implicit methods that aim to extract vectorized semantic representations from papers [Meijer *et al.*, 2021] are hard to parse at the human level, let alone operate on such representations.

Recent breakthroughs in large language models (LLMs), particularly generalized pre-trained transformer techniques [Brown *et al.*, 2020; OpenAI *et al.*, 2023], have demonstrated exceptional zero-shot/few-shot capabilities across various downstream tasks and have shown broad domain knowledge coverage [Bubeck *et al.*, 2023]. The synergy between LLMs and knowledge graphs constitutes an active area of research. LLMs have shown reasonable performance in tasks such as entity identification for knowledge graph construction, and their capabilities can be significantly enhanced when coupled with knowledge graphs as external knowledge sources [Pan *et al.*, 2023; Zhu *et al.*, 2023].

Armed with this advancement, in this study, we explore the possibility of using LLMs as a bridging tool by distilling concepts from research papers in astronomy and astrophysics and constructing knowledge graphs to study their relationships and co-evolution over time. To the best of our knowledge, this is the first time an LLM-based knowledge graph has been constructed for astrophysics. The combination of the LLM-extracted concepts with our proposed citation-reference-based relevance allows us to quantitatively analyze cross-domain interactions over time and the co-evolution of subfields in astronomy.

This paper is organized as follows: In Section 2, we outline the dataset used for this study. Section 3 details the methodologies employed, including knowledge graph construction with large language model agents and the citation-reference-based relevance to quantify the interconnection between different concepts. We present our findings in Section 4, including a case study focusing on how numerical simulations were gradually adopted by the astronomical community, and by extension, quantifying the current impact of machine learning in astronomy. We discuss and conclude in Section 5.

## 2 Literature in Astronomical Research

This study employs a dataset of $297,807$ arXiv papers in the fields of astronomy and astrophysics, collected from 1993 to 2024 and sourced from the NASA Astrophysics Data System (NASA/ADS) [Accomazzi, 2024]. Astrophysics is known to be a field where the vast majority of publications are on arXiv and easily searchable on ADS. Therefore, the number of arXiv papers here comprises a close-to-complete collection of literature that was published in the field.

We downloaded all PDFs from arXiv and performed OCR with Nougat [Blecher *et al.*, 2023]. Through human inspection, we found that Nougat did a great transcription of the data with minimal failure. The same set of data was currently used to train various specialized LLMs in astronomy (Pan et al., in prep., Arora et al,, in prep.), following AstroL-LaMA and AstroLLaMA-Chat [Dung Nguyen *et al.*, 2023; Perkowski *et al.*, 2024], and auxiliary minor mistakes were identified and cleaned up during those iterations.

A key component of this paper is understanding the relation of concepts, as viewed by the research community, through the citation relation within the existing literature. The fact that NASA/ADS oversees a close to complete literature makes astronomy one of the well-curated fields to explore this study. We further extract the citation-reference relation for the entire corpus using the NASA/ADS API[1] to quantify the interaction among various scientific concepts during their co-evolution.

## 3 Constructing a Knowledge Graph for Astronomy

Constructing a knowledge graph between concepts in astrophysics requires two essential components: extracting the concepts in astronomical literature through large language model agents, and determining the strength of interconnectivity between concepts through the underlying relationships between paper citations. In this section, we explore these components in more detail.

### 3.1 Concept Extraction with Large Language Models

The key challenges in distilling concepts from publications using large language models are twofold. Firstly, LLM agents may generate hallucinations, producing lists of concepts that deviate from the expectations of human experts. Secondly, even when the concepts are accurately distilled, the models may yield concepts that are either too detailed, overly broad, or merely synonymous with each other, thereby diminishing the practical relevance of understanding their interrelationships. To address these challenges, we employ a multi-agent system in this study, as shown in Figure 1. This system consists of three parts: (a) extraction of concepts from astronomical publications; (b) nearest neighbor search of the concepts; and (c) merging of the concepts. This iterative approach enables control over the granularity of the knowledge graph, tailoring it to our purpose.

In this study, we focus on extracting key concepts from the titles and abstracts of astronomical publications to minimize computational cost. In astronomy, the abstract often encapsulates the essential information, including scientific motivation, methods, and data sources. The abstracts from the 300,000 papers amount to a total of approximately 2 billion tokens. To efficiently handle this large-scale data while maintaining cost-effectiveness, we leverage open-source large language models for concept extraction. Specifically, we employ MISTRAL-7B-INSTRUCT-V0.2[2] [Jiang *et al.*, 2023] as our inference model and JINA-EMBEDDINGS-V2-BASE-EN[3] [Günther *et al.*, 2023] for text embedding.

---

[1] https://ui.adsabs.harvard.edu/help/api/

[2] https://huggingface.co/mistralai/Mistral-7B-Instruct-v0.2

[3] https://huggingface.co/jinaai/jina-embeddings-v2-base-en

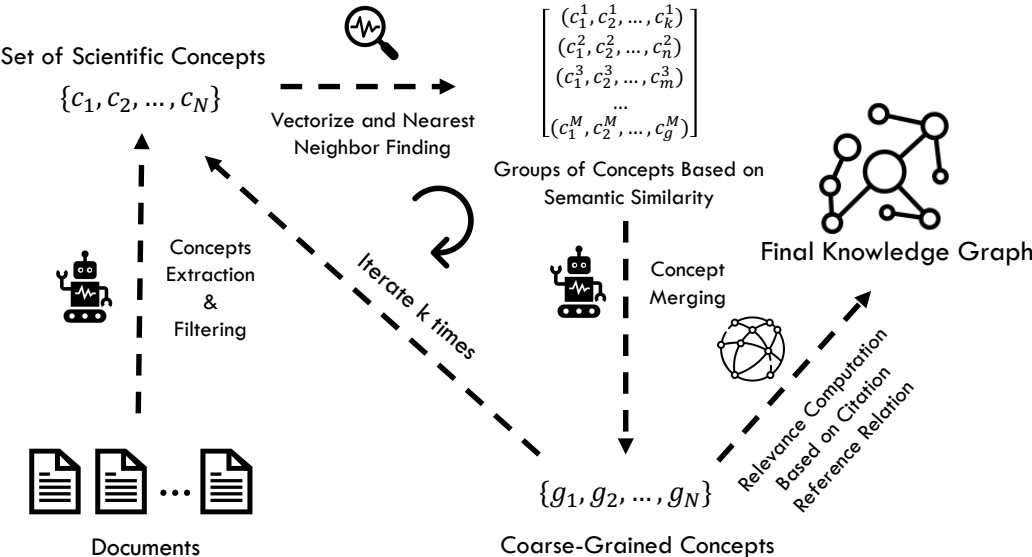

Figure 1: Schematic plot outlining the knowledge graph construction using large language model agents. The extraction of concepts comprises three main phases: (1) Concept Extraction, where agents construct scientific concepts from documents; (2) Vectorization and Nearest Neighbor Finding, in which concepts are vectorized and grouped by semantic similarity; (3) Concept Merging, where similar concepts are combined to form a more coarse-grained structures. The connections between concepts are then defined by citation-reference relevance as detailed in Section 3.2, with concepts involved in more citation-reference pairs assigned a higher relevance.

**Concept Extraction:** The first agent is prompted to extract a preliminary set of scientific concepts from the abstracts and titles[4]. While most of these concepts appear to be valid, some of them seem to be hallucinations that are not pertinent to astronomy, such as "misleading result" and "maternal entity in astronomy". To address this issue, a secondary LLM agent is deployed to explain and clarify each term, ensuring the removal of ambiguities and allowing only scientifically valid concepts to proceed. In this clarifying step, we utilize the entire document as an additional source enhanced by retrieval augmented generation to assist our agent in accurately understanding the meanings of various scientific terminologies. The validated scientific concepts are denoted as $\{c_1, c_2, \ldots, c_N\}$.

**Vectorize and Nearest Neighbor Finding:** Once the concepts are extracted and validated, they are transformed into vector representations using the text-embedding models, enabling the accurate computation of similarity measures. We group the concepts based on the cosine similarity of their corresponding vector representations into $M$ clusters, represented as $\{\{c_j^i, j = 1, \ldots, k_i\}, i = 1, \ldots, M\}$. The number of elements in each cluster, $k_i$, is adaptively determined based on a predefined cosine similarity threshold among the elements within the cluster. In this study, we set the threshold at 0.85, striking a balance between the granularity of concepts and the computational feasibility of the subsequent steps.

---

[4]All code and prompts will be made public after review.

**Concept Merging:** Finally, the final agent merges these grouped concepts by analyzing clusters of semantically similar concepts and distilling them into more general, unified entities. For example, the concepts "X-Shooter spectra", "Saturn's transmission spectrum," and "Keck LRIS spectrograph" were combined into the broader concept of "spectrograph". This merging simplifies the structure of the knowledge graph, reducing redundancy. Furthermore, a coarser knowledge graph improves the readability of the visualization.

We iterate the neighbour finding and merging steps three times, gradually coarsening the collection of concepts from 1,057,280, 164,352, and finally 24,797 concepts, respectively. We found, through domain expert evaluation that, the granularity of the concepts after three iterations is appropriate, with sufficient concepts covering the broad range of topics explored and methods employed in the literature, but with enough fine-grained level to understand the subtle evolution of the field in astrophysics. Some of the final concepts include the commonly known concepts such as "dark matter", "inflation", and etc. On average, each paper consists of $\sim 10$ concepts.

## 3.2 Determining Concept Relevance

Upon defining the concepts, perhaps more critical is to determine, quantitatively, how strongly two concepts are relevant. The relevancy of two concepts is certainly subjective—concepts that were deemed irrelevant at a certain point in time by the domain expert community might gradually become relevant over time. However, such temporal evolution

is exactly what we are after to understand the shift of knowledge over time.

To gauge how two concepts are perceived as relevant by the community at a fixed point in time, the citation-reference relationships between articles become a natural annotated link between the concepts. In the following, we will define based on the probability with which a pair of concepts appears simultaneously in a certain article and its neighboring documents that have a citation-reference relationship, the proximity of the two concepts. This metric between concepts is inspired by the process by which researchers randomly sample through the network of articles from one concept to another. If the researcher can find another new concept from the parent concept that they were originally interested in by searching through the direct citation relation from the paper which contains the parent concept, and this leads the researcher to another paper with a new concept, the two concepts are deemed close. However, if the two concepts can only be found through a small subset of papers of the parent concepts and their citations or references, then the two concepts are deemed further apart at that point in time. We emphasize that while the linkage (and here, the hypothetical "search") is done through the domain of the published literature, the knowledge graph is constructed at the level of the extracted concepts.

More formally, let the final set of concepts be denoted as $C : \{c_1, c_2, \ldots, c_n\}$, identified using large-language model-based agents as outlined in Section 3.1. Let these concepts be associated with a corpus of academic papers, $N : \{n_1, n_2, \ldots, n_k\}$, and a set of citation-reference relationships $L : \{(n_a, n_b) | n_a, n_b \in N, \exists n_a \rightarrow n_b\}$, where $n_a \rightarrow n_b$ signifies that paper $n_a$ cites paper $n_b$. To explore the propagation of a concept $c_\alpha$ within this network, we define the probability of encountering another concept $c_\beta$ starting from a specific paper $n_k$ that discusses $c_\alpha$. This probability, denoted as $p_{\alpha \rightarrow \beta | n_k}$, is formulated as:

$$p_{\alpha \rightarrow \beta | n_k} = \frac{N_\beta}{|S(n_k, L, \beta)|}. \tag{1}$$

The set $S(n_k, L, \beta)$ is defined through an iterative process starting with the initial paper set $n_k$ (denoted as $S_0$). In each iteration, we expand the set by including papers that are directly cited by any paper in the current set and have not been included in previous sets. Formally, if $S_{n-1}$ is the set of papers at iteration $n-1$, then $S_n = S_{n-1} \cup \{n_e | (n_s, n_e) \in L, n_s \in S_{n-1}, n_e \notin S_{n-1}\}$. The iteration continues until at least one paper in the current set contains concept $c_\beta$, at which point we denote the final set as $S_T$ and set $S_T = S(n_k, L, \beta)$. The number of papers containing $c_\beta$ within $S(n_k, L, \beta)$ is set to be $N_\beta$.

Typically, the growth of the sets follows a pattern where $|S_0| = 1$, $|S_1| \sim 10^2$, and $|S_2| \sim 10^4$ in our experiments. This means that if the concepts cannot be found directly from a direct citation from the original paper that contains the parent concept, the number of papers "needed to be read", i.e., $|S|$, will drastically reduce the relevance of the two concepts. Nonetheless, if the concepts are very prevalent, after a certain level of search, the numerator $N_\beta$ would then offset the volume of search.

As this probability pertains to just a specific paper containing concept $c_\alpha$, the probability of transitioning from concept $c_\alpha$ to $c_\beta$, for all the papers $S_\alpha$ that contain $c_\alpha$, would then be the expectation averaging over all papers in $S_\alpha$, or,

$$p_{\alpha \rightarrow \beta} = \frac{1}{|S_\alpha|} \sum_{n_k \in S_\alpha} p_{\alpha \rightarrow \beta | n_k} \tag{2}$$

The above equation computes the average probability of moving from $c_\alpha$ to $c_\beta$ across all papers that contain $c_\alpha$. To assess the bidirectional relevance of concepts $c_\alpha$ and $c_\beta$, and we will assume that the order of transition between two concepts is not relevant, we define the citation-reference relevance between them as the geometric average of the probabilities of transitioning in both directions:

$$p_{\alpha, \beta} = (p_{\alpha \rightarrow \beta} \cdot p_{\beta \rightarrow \alpha})^{1/2} \tag{3}$$

Finally, the transition probability attains the following trivial properties: (1) $p_{\alpha, \beta} \leq 1, \forall c_\alpha, c_\beta \in C$; (2) $p_{\alpha, \alpha} \equiv 1, \forall c_\alpha \in C$; and (3) $p_{\alpha, \beta} = p_{\beta, \alpha}, \forall c_\alpha, c_\beta \in C$. These properties ensure that the relevance metric is well-defined and consistent, providing a foundation for analyzing the relationships between concepts in the knowledge graph.

### 3.3 From Concept Relevance to Knowledge Graph

From the relevance defined as $p_{\alpha, \beta}$ above, which serves as a robust metric for the link strength between two nodes, we can visualize the knowledge as a force-directed graph. A force-directed graph [Kobourov, 2012; Bannister *et al.*, 2012], alternatively known as a spring-embedder or force-based layout, serves as a visual tool designed to illustrate relational data within network graphs. This method leverages simulation techniques inspired by physical systems, arranging nodes—which symbolize entities or concepts—and links—which depict the relationships or connections between these nodes—in an aesthetically coherent and insightful layout. These graphs utilize the concept of attraction and repulsion forces to strategically distribute nodes.

By iteratively updating the positions of nodes based on these attraction and repulsion forces, the force-directed graph algorithm converges to a layout that minimizes the overall energy of the system. This results in an informative 3D representation of the knowledge graph, where closely related concepts are automatically positioned near each other, enhancing the visibility of the density and connectivity within the graph. The capacity of force-directed graphs to dynamically represent complex relational data makes them particularly suitable for visualizing the knowledge graph.

In our context, the link strength between two nodes (concepts) is set to their citation-reference relevance, $p_{\alpha, \beta}$. Concepts with higher relevance will attract each other more strongly [Cheong *et al.*, 2021], causing them to be positioned closer together in the visualized graph. Conversely, the repulsion force is applied between all pairs of nodes, ensuring that they remain adequately spaced to prevent overlap and maintain clear visual separation. By leveraging the citation-reference relevance as the link strength between concepts, we can create a graph that intuitively conveys the relationships and clustering of ideas within the astronomical literature.

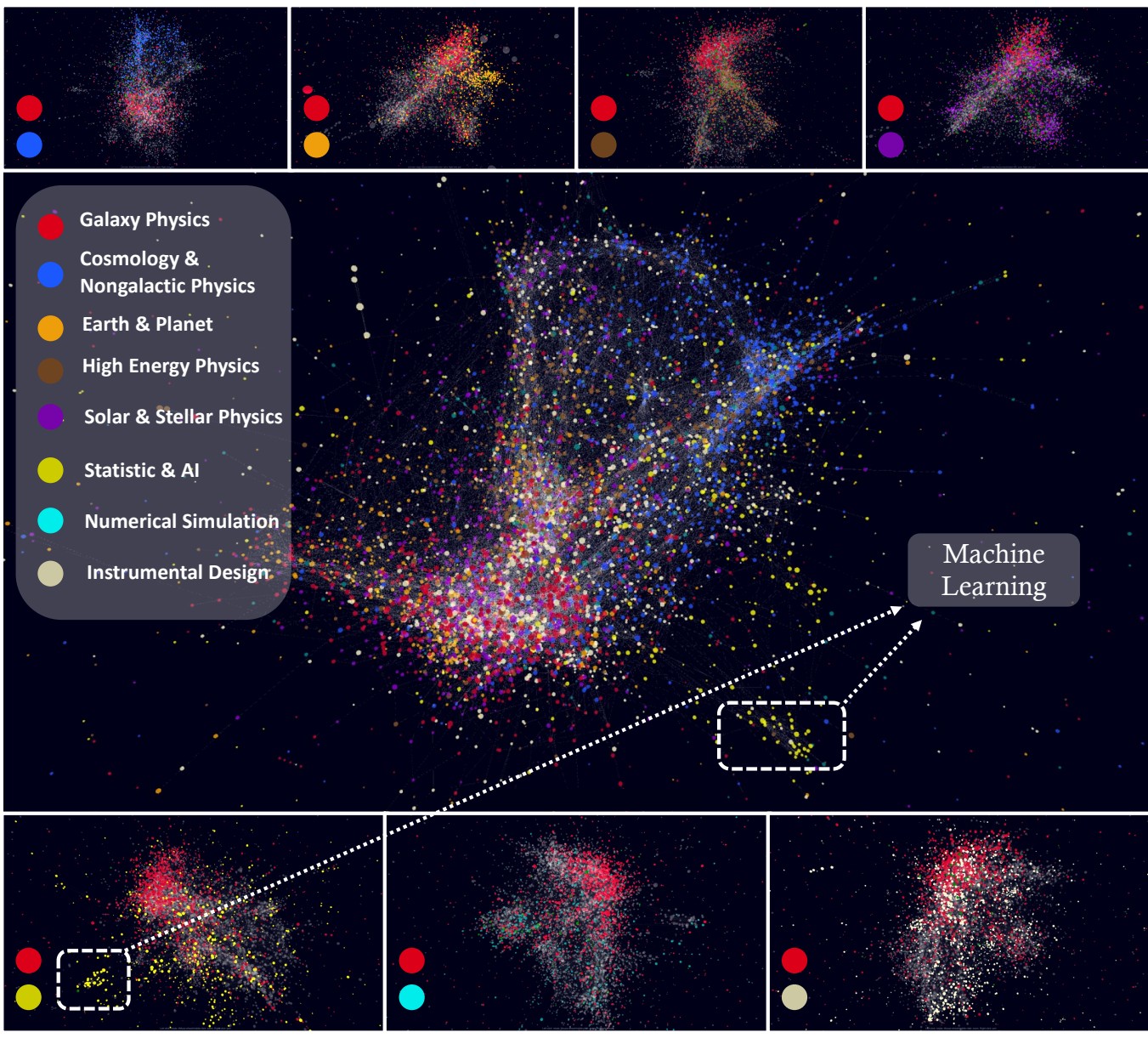

Figure 2: Visualization of a knowledge graph of 24,939 concepts, constructed from 297,807 astronomical research papers. Only concepts appearing in more than 20 papers and links with a link strength greater than 0.001 are displayed. Each concept is categorized into one of the following domains: (A) Galaxy Physics, (B) Cosmology & Nongalactic Physics, (C) Earth & Planetary Science, (D) High Energy Astrophysics, (E) Solar & Stellar Physics, (F) Statistics & AI, (G) Numerical Simulation, or (H) Instrumental Design. In the upper panels, we show connections between galaxy physics and other scientific domains. In the lower panel, we highlight the concepts from simulation, statistics, and observational instruments and their respective locations with respect to galaxy physics. Unsurprisingly, the technological concepts are generally more globally spread, as the same techniques can have wide implications for a broad range of topics in astronomy. Machine learning techniques are still at the periphery of the knowledge graph, suggesting that their integration in astronomy is still in its early stages. The interactive version of the knowledge graph is made publicly available after review.

## 4 Intersection between Technological Advancement and Scientific Discovery

Our knowledge graph consists of 24,939 concepts, extracted from 297,807 astronomical research papers, with 339,983,272 interconnections. The visualization of the knowledge graph as a force-directed graph is shown in Figure 2. The filamentous structure shown in the knowledge graph demonstrates the close interconnections across various subdomains within astronomical research.

For clarity, we only display concepts that appear in at least 20 papers and consider only those links with a citation-reference relevance $p_{\alpha,\beta} > 0.001$. This leads to 9,367 nodes and 32,494 links for the visualization. We set the size of the nodes to be proportional to the logarithm of their frequency of occurrence in the papers.

In the visualization, we further categorize all the concepts into scientific concepts, following the categorization of astrophysics on arXiv[5], namely Astrophysics of Galaxies,[6] Cosmology and Nongalactic Astrophysics,[7] Earth and Planetary Astrophysics,[8] High Energy Astrophysics,[9] and Solar and Stellar Astrophysics,[10]. As we aim to understand how concepts in technological advancement propel scientific discoveries, we further define another three classes of "technological" domains, which we identify as Statistics and Machine Learning, Numerical Simulation, and Instrumental Design. The classifications below are conducted using GPT-4[11].

Figure 2 illustrates how relevant concepts cluster within the same domain and how different domains interconnect. The upper panels demonstrate how the different scientific clusters interact with each other. For instance, galaxy physics, as anticipated, connects with both the largest scales in astronomical research, such as cosmology and general relativity, and the smaller scales, including stellar physics and planetary physics. The lower panel shows how the technological concepts are embedded within the scientific concepts, including numerical simulations, statistics, machine learning, and instrumental design. The technological concepts are generally distributed more globally in the knowledge graph, demonstrating their omnipresence in different subfields.

Interestingly, as shown in the figure, despite the booming interest and popularity, machine learning techniques, particularly deep learning, are situated only at the peripheral region

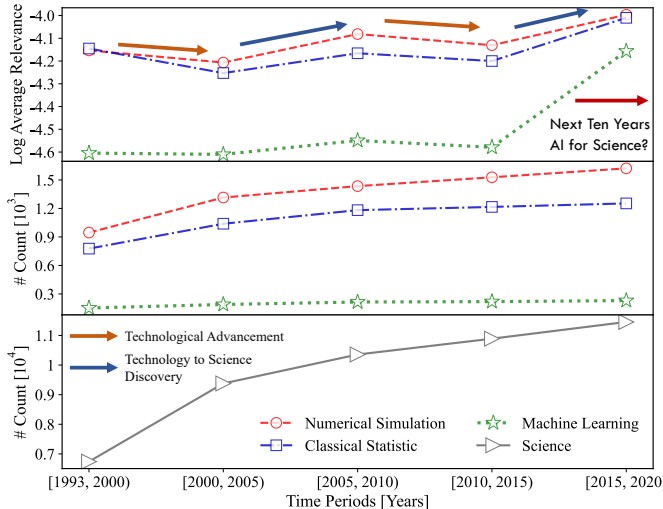

Figure 3: The citation-reference relevance for five distinct time periods to investigate the temporal integration of technological techniques into scientific research. The middle and lower panels illustrate a consistent increase in the count of concepts, both in terms of scientific concepts (bottom panel) and technical concepts (middle panel). The upper panel shows the total cross-linkage between individual technical domains and scientific concepts, with lower values indicating stronger adoption. The upper panel reveals a two-phase evolution, with an observed latency of approximately five years. The two phases signify the period of development and introduction of new techniques in astronomy and their subsequent adoption by the community (see text for details). While still modest, machine learning has begun to reach integration levels comparable to those of numerical simulations seen two decades earlier.

of the knowledge graph. This suggests that machine learning techniques are not yet fully integrated into the astronomical research community, at least from the citation-reference point of view. We will provide a more quantitative comparison of this observation in the following section.

### 4.1 Numerical Simulations in Astronomy

To demonstrate how technological advancement drives scientific discovery, we will study in more depth the impact of numerical simulations on astronomy. In modern-day astronomical research, numerical simulation has become an indispensable tool. However, this was not always the case. The scientific community experienced a gradual transition from focusing primarily on theoretical deduction and analytical formulas to modeling complex phenomena through numerical simulations.

To understand this transition, we assess the average relevance between numerical simulations and scientific concepts across various time periods. We divided the dataset into five time periods from 1993 to 2020. In each time period, we recalculate the citation-reference relevance using the papers published within that specific timeframe.

As shown in the bottom panel of Figure 3, unsurprisingly, the number of "scientific concepts" has surged over time. Complementary to these scientific concepts, we also see that the number of technical concepts has surged alongside, espe-

---

[5]https://arxiv.org/archive/astro-ph

[6]Astrophysics of Galaxies focuses on phenomena related to galaxies and the Milky Way, including star clusters, interstellar medium, galactic structure, formation, dynamics, and active galactic nuclei.

[7]Cosmology and Nongalactic Astrophysics covers the early universe's phenomenology, cosmic microwave background, dark matter, cosmic strings, and the large-scale structure of the universe.

[8]Earth and Planetary Astrophysics studies deal with the interplanetary medium, planetary physics, extrasolar planets, and the formation of the solar system.

[9]High Energy Astrophysics explores cosmic ray production, gamma ray astronomy, supernovae, neutron stars, and black holes.

[10]Solar and Stellar Astrophysics pertains to the investigation of white dwarfs, star formation, stellar evolution, and helioseismology.

[11]https://openai.com/index/gpt-4/

cially in terms of numerical simulations and statistical methods, which are shown as red and blue lines in the middle panel. On the other hand, despite the interest in the field, the number of concepts in machine learning in the astronomical literature, as shown by the green line, is still an order of magnitude lagging behind these other well-developed technological concepts.

Perhaps more interesting is showing the weighted "intersection" between the scientific concepts and the technical concepts, which is shown in the top panels. The top panel shows the weighted "linkage" among all the scientific concepts with the specific technical domain. If the new methods are well-adopted in the astronomical community and advance scientific discovery, we should see an improvement in the average citation-reference linkage (large values in the top panel). Viewed this way, there is a clear two-phase evolution with the gradient of the integration oscillating positively (blue arrow) and negatively (red arrow).

This is perhaps not surprising. For any technological advancement, it might once be proposed with many technically focused papers written; however, the citation-reference relation is mostly limited to the "technologists," leading to a dilution of the cross-correlation, which is shown by the red arrow. For example, during the period of 1993-2000, there have been many works focusing on the development of N-body simulation techniques [?; Romeo *et al.*, 2004; Springel, 2005]. Yet, the integration remains marginal. However, from 2000 onward, the astronomical community began to embrace N-body simulations to resolve scientific questions [Paz *et al.*, 2006; Peñarrubia *et al.*, 2006; Zhou and Lin, 2007], resulting in a increase in citation-reference relevance during this time. A similar two-phase pattern is observed from [2010, 2015] to [2015, 2020], during which time hydrodynamical simulations developed [Genel *et al.*, 2014; Carlesi *et al.*, 2014b; Carlesi *et al.*, 2014a] and gradually gained acceptance [McAlpine *et al.*, 2016; Pillepich *et al.*, 2018] within the community. The delay between the development of new technologies and their impact on scientific discovery spans approximately five years.

### 4.2 Machine Learning in Astrophysics

The revelation of the two-phase adoption in numerical simulations leads to the possibility of better quantifying the integration of machine learning in astronomy. In recent years, we have seen a booming interest in AI and its applications in science. As modern-day astronomy is driven by big data, with billions of sources routinely being surveyed, it is not surprising that astronomy has also seen a drastic integration of AI to advance data processing and analysis [Baron, 2019].

Figure 4 shows the average cross-domain linkage, as defined in the top panel of Figure 3, but between the concepts in machine learning and the five scientific domains. In terms of the application of machine learning in astronomy, Cosmology & Nongalactic Astrophysics takes the lead, as it benefits from machine learning's capacity to manage complex, large data sets from simulations and surveys [Villaescusa-Navarro *et al.*, 2021b; Villaescusa-Navarro *et al.*, 2021a; Sun *et al.*, 2023b]. This is followed by Galaxy Physics, which leverages ML for tasks like photometric redshift pre-

diction [Sun *et al.*, 2023a] and galactic morphology classification [Robertson *et al.*, 2023]. Solar and Stellar Physics have also shown promise in emulating and analyzing stellar spectra [Ting *et al.*, 2019]. High Energy Astrophysics and Earth & Planetary Astrophysics have been slower to adopt ML.

But is machine learning now well-adopted in astronomical research? Figures 2 and 3 paint an interesting picture. On the one hand, the top panel of Figure 3 shows that there has been a rapid increase in the cross-science-and-AI citation-reference relevance, demonstrating a huge interest among the community. For instance, the scientific-technology score remains flat and low before 2015, signifying that despite a history of AI in astronomy—such as the use of neural networks for galaxy morphology classification as early as 1992 [Storrie-Lombardi *et al.*, 1992]—its impact remained minimal until the surge in popularity of deep learning post-2015.

Yet, at the same time, even currently, Figure 2 shows that most of these concepts still occupy a peripheral position in the knowledge graph. This suggests that, from a citation-reference relevance perspective, such concepts are still considered niche within the broader scientific community. This is perhaps not too surprising because, compared to the deep integration of numerical simulations, quantitatively, the cross-linkage score of machine learning with astronomy remains only at the level that numerical simulations and traditional statistics were twenty years ago.

Perhaps what is strikingly lacking is that the number of machine learning concepts in the astronomical literature remains an order of magnitude smaller than that of numerical simulations, as shown in the middle panel of Figure 3. This might imply that the machine learning techniques widely adopted in astronomy, even at present, remain some of the more classical techniques, such as linear regression and random forests [Nyheim *et al.*, 2024]. The rapid adoption of "existing" techniques, while encouraging, might also signify a bigger underlying problem of lack of innovation in applying AI to astronomy. However, if the two-phase evolution applies, we should expect that in the coming years, there will be more novel deep learning techniques introduced before they are gradually adopted by the community.

## 5 Discussions and Conclusions

A quantitative study of the evolution of concepts and their interconnections would not be possible without modern-day LLMs, partly due to the large amount of arduous work required to manually label, extract concepts, and classify topics, which can be easily done with minimal computing resources in our case. Even when manual extraction is possible, the taxonomy of a scientific field is often limited—tailored to provide vague contours of the domain, e.g., for publication purposes, rather than a deep and more fine-grained differentiation of the knowledge embedded in the field.

In this study, we construct, to the best of our knowledge, the first LLM-based knowledge graph in the domain of astronomy and astrophysics. The knowledge graph comprises 24,939 concepts extracted through a careful iterative process with LLMs from 297,807 papers. We design a relevance metric defined through the citation-reference relations in the as-

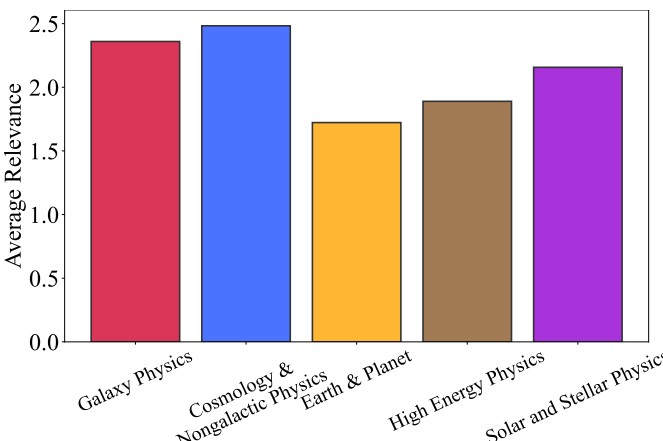

Figure 4: Integration of machine learning in different subfields of astronomy. The integration is defined as the average cross-domain linkage similar to the top panel of Figure 3. Cosmology and Nongalactic Astrophysics currently lead the application of machine learning in astronomy, followed by Galaxy Physics and Solar & Stellar Physics. The adoption of machine learning concepts in Earth & Planetary Physics and High Energy Astrophysics still lags behind.

tronomical literature to understand the relations as well as the temporal evolution between different concepts. The relevance metric follows the intuition of how humans search for new concepts by quantifying the degree of separation in the citation network as well as the prevalence of the concepts in the field. The relevance is then applied as the linkage strength of the force-directed graph to construct the knowledge graph, allowing us to visualize the knowledge in the field in detail.

Based on this knowledge graph, we evaluate the temporal evolution of the relevance of numerical simulations and machine learning in astronomical research. We showed that while numerical simulations are routinely adopted in modern-day astronomy, the concepts related to them have gone through a long process of gradually being integrated into and accepted by the community. We also found that the integration of numerical simulation into scientific discovery shows a two-phase process, in which a five-year latency can be observed between the development of techniques, where the relevance of the techniques and the science might temporarily diminish, followed by the flourishing period, where the methods mature and are widely applied to astronomical research. We also found that the same trend can be found in classical statistical analysis.

By the same metric, we found that, despite much of the interest and the booming field of deep learning, the impact of deep learning in astronomy remains marginal. While there is a drastic increase in the technique-science cross-referencing, quantitatively, the referencing remains at a level that we observed for numerical simulations about two decades ago. Furthermore, the number of machine learning concepts introduced in astronomy remains an order of magnitude smaller than that of numerical simulations and classical statistical methods, which might imply that the current rapid increase in relevance is driven mainly by the adoption of established

machine learning techniques from decades ago. Nonetheless, if the two-phase transition applies, we expect more innovative techniques will be gradually introduced. In fact, in recent years, we have seen many more modern-day techniques, both in terms of flow-based and score-based generative models [De Santi *et al.*, 2024; Zhao *et al.*, 2023], being introduced, as well as, like this study, the application of LLMs in astronomical research [Dung Nguyen *et al.*, 2023; Perkowski *et al.*, 2024]. The metric introduced here will be able to continue monitoring this process.

This study primarily aims to show a proof of concept, using LLM-based Knowledge Graph to quantifiably understand the evolution of astronomical research. As such our study certainly has much room for improvement. For instance, proper robust extraction of scientific concepts from literature heavily relies on the alignment between the agents and the researchers' perception. In our study, the concepts are autonomously extracted through the LLM agent, with the granularity of the concepts optimized through merging and pruning. Such an LLM agent can certainly benefit from a subset of high-quality annotated data and comparison with existing hierarchical taxonomies. The process of concept pruning and merging is also somewhat crude, involving vectorizing the concepts and performing a cosine similarity search. A better method would involve further comparing these concepts, utilizing the capabilities of large language models for more detailed concept differentiation and pruning.

In a nutshell, our study demonstrates the potential of LLM-based knowledge graphs in uncovering the intricate relationships and evolution of astronomical research. By providing a quantitative framework for analyzing the integration of new technologies and methodologies, this approach opens up new avenues for understanding the dynamics of interdisciplinary research and the factors that drive scientific progress, in astronomy and beyond.

## Ethical Statement

In this study, we construct a knowledge graph by extracting concepts from the astronomical literature available on the arXiv preprint server. Our work aims to advance the understanding of the evolution and interconnections of scientific concepts within the field of astronomy. We emphasize that our study does not involve the direct reproduction or distribution of the original literature itself. Instead, we focus on distilling and analyzing the key concepts present in the existing body of work.

To ensure ethical compliance and respect for intellectual property rights, we will only release the extracted concepts and their relationships, without sharing or reproducing the original text or any substantial portions of the literature. This approach minimizes the risk of copyright infringement and maintains the integrity of the original authors' works.

Furthermore, the field of astronomical research generally operates under an open-sky policy, which promotes collaboration, transparency, and the free exchange of scientific knowledge. This policy aligns with our research objectives and mitigates potential ethical or monetary disputes arising from our work. Our goal is to provide insights that benefit the

astronomical community and contribute to the advancement of scientific understanding.

## Acknowledgments

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
