# OpenReview forum: "Knowledge Graph in Astronomical Research with Large Language Models: Quantifying Driving Forces in Interdisciplinary Scientific Discovery"
_ijcai.org/IJCAI/2024/Workshop/AI4Research — AI4Research 2024_

### Official Review · Reviewer_aVKF · 2024-05-31
**Marginally below acceptance threshold (weak reject)**

**Rating:** 5
**Confidence:** 4

**Review:**

The paper proposed a method that uses LLMs as agents to extract the concept from astronomical papers. The first LLM agent is designed to extract concepts from the title and abstract, and the second one is used to analyze the whole paper to avoid hallucination. The last LLM agent is designed to merge the concept. There are some steps in the method that are not clear and strange to me.

The advantages of the paper are as follows:
1. The authors leverage LLMs to build the knowledge graph in the astronomical area, which is nice and can reduce the human resource consumption for labeling. The knowledge graph can be used to mine more information in the astronomical area.
2. the description is clear and easy to follow up.

However, there are some points in the paper that are not that clear and strange.
1. in the concept extraction section, authors proposed using two LLM agents, one is for concept extraction and another one is used for reducing the hallucination based on the entire document. This is strange to me. If one agent is built based on the entire document, why not use only one agent to extract the concepts based on the entire document? Or, assume this agent is only used for the hallucination concepts, how do you identify the hallucination concepts in this huge amount of concepts? This section is not clear.
2. Is the agent actually needed for the concept merge? I saw that authors use vectors and similarity to find similar concepts. In my opinion, a regular language model can also be used to conclude the concepts, which will reduce the cost.
3. I think the authors can show some prompt examples in the paper so that we can understand how to get those concepts.
4. Some works also use LLM to extract the entities/concepts from the astronomical area.
5. Typo in the paper. One citation was missed in section 4.1.

---

### Official Review · Reviewer_5Gvy · 2024-06-03
**This study leverages large language models to extract 24,939 key concepts from 297,807 astronomy publications (1993-2024). By constructing a knowledge graph based on citation relationships, it measures the impact of concepts over time, particularly numerical simulations and AI. This method successfully quantifies the integration of interdisciplinary research, demonstrating its potential to understand scientific evolution and breakthroughs.**

**Rating:** 7
**Confidence:** 3

**Review:**

**Quality**: The study is well-structured, leveraging advanced large language models (LLMs) and knowledge graphs to achieve its objectives. The extraction of concepts from an extensive dataset of astronomical publications and the subsequent construction of a knowledge graph demonstrate a high level of technical execution. However, some aspects of the methodology, such as concept pruning and merging, could be refined for greater accuracy. Reliance on autonomous extraction by LLMs without high-quality annotated data might miss nuances understood by human experts.

**Clarity**: The work is clearly written, with a logical flow that makes it accessible to readers with a background in the subject. The explanation of the two-phase integration process for numerical simulations and the comparison with deep learning trends are particularly clear.

**Originality**: This study stands out for its innovative application of LLMs to extract and analyze research concepts over an extensive period. The approach of constructing a knowledge graph to measure the integration of new technologies in astronomy is novel and provides a new quantitative framework for understanding interdisciplinary research dynamics.

**Significance**: The findings have significant implications for the field of astronomy and beyond. By quantifying the integration of numerical simulations and deep learning, the study provides valuable insights into how new technologies influence scientific progress. The methodology has the potential for broad application across various scientific domains, making it a substantial contribution to the literature on interdisciplinary research.

---

### Decision · Program_Chairs · 2024-06-03

Accept